# Transcatheter Aortic Valve Implantation for Pure Native Aortic Regurgitation: The Last Frontier

**DOI:** 10.3390/jcm11175181

**Published:** 2022-09-01

**Authors:** Ana Paula Tagliari, Rodrigo Petersen Saadi, Eduardo Keller Saadi

**Affiliations:** 1Post Graduate Program in Cardiology and Cardiovascular Science, Federal University of Rio Grande do Sul, Porto Alegre 90035-002, Brazil; 2Cardiovascular Surgery Department, Hospital São Lucas da PUC-RS, Porto Alegre 90619-900, Brazil; 3Cardiovascular Surgery Department, Hospital Mãe de Deus, Porto Alegre 90880-0481, Brazil; 4Cardiovascular Surgery Department, Hospital de Clínicas de Porto Alegre, Porto Alegre 90035-903, Brazil

**Keywords:** transcatheter aortic valve implantation, aortic regurgitation, self-expanding devices, balloon-expandable devices

## Abstract

Transcatheter aortic valve implantation (TAVI) to treat patients with severe symptomatic aortic stenosis is a well-established procedure. Even though cases series have reported TAVI use in high-risk patients with pure native aortic regurgitation, this is still considered an off-label intervention, especially when the aortic annulus dimensions are beyond the recommended by prosthesis manufacturers. Herein, we provide an updated review regarding the transcatheter treatment of pure native aortic regurgitation and illustrate this issue by presenting a clinical case of a patient with pure aortic regurgitation and a large aortic annulus who received a self-expanding non-dedicated transcatheter heart valve.

## 1. Introduction

Transcatheter aortic valve implantation (TAVI) evolved from a therapy reserved for inoperable [1] and high-risk patients [2,3] to a well-established and safe procedure indicated even for intermediate [4,5] and low-risk populations [6,7]. Regardless of the patient risk class, TAVI was conceptualized to address severe aortic valve stenosis since a certain amount of annulus and leaflets calcification is presumably essential to anchor the valve-mounted stent-frame. Therefore, TAVI in pure native aortic regurgitation (PNAR) using conventional transcatheter heart valves (THV) is considered an off-label intervention.

In spite of this, the results of the PNAR transcatheter approach with non-dedicated devices, such as the self-expanding CoreValve/Evolut (Medtronic, Minneapolis, Minnesota, USA) or the balloon-expandable SAPIEN (Edwards Lifesciences LLC, Irvine, CA, USA) have been reported in several case reports and in small clinical studies [8,9,10].

To overcome the challenges inherent to TAVI performance in PNAR, two new THV, with a technology dedicated to approaching aortic regurgitation (AR), were developed, the JenaValve™ (JenaValve Technology GmbH, Munich, Germany) and the J-valve™ (JC Medical, Inc. Burlingame, CA, USA and Suzhou, China) [11,12,13].

Herein, we provide an updated review regarding the transcatheter treatment of PNAR and illustrate this issue by presenting a challenging case of a patient with PNAR and a large aortic annulus who received a self-expanding non-dedicated THV.

## 2. Clinical Vignette

A 75-year-old male patient with a previous open mitral valve replacement complicated by mediastinitis, multiple sternal reinterventions, and chest closure by secondary intention 4 years ago was admitted to our hospital presenting dyspnea, classified as New York Heart Association (NYHA) functional class IV, and peripheral edema. 

Transthoracic and transesophageal echocardiograms showed a PNAR with an eccentric regurgitant jet (regurgitant volume 54 mL/beat, regurgitant fraction 50%, diastolic flow reversal in the descending aorta, end-diastolic velocity 21 cm/s). Besides, he had left ventricle (LV) dysfunction with a LV ejection fraction (LVEF) of 47% (Figure 1a,b). 

A computed tomography (CT) angiography revealed a huge aortic annulus (aortic annulus area 780 mm^2^, aortic annulus perimeter 99 mm) and no annulus or leaflets calcification (Figure 2a,b). Maximal ascending aorta diameter was 32 mm. The coronary arteries and the arterial iliofemoral system anatomy were considered suitable for TAVI. 

After institutional heart team discussion and agreement of contraindication to redo cardiac surgery (hostile chest, EuroScore II 7.14%, STS score 4%), we decided to perform a TAVI using the biggest self-expanding bioprosthesis available in our center (Evolut R 34 mm). 

The procedure was performed under general anesthesia and transesophageal echocardiogram guidance. Two pigtail catheters were positioned in the Sinus of Valsalva to work as an annulus landmark (Figure 3a). The THV was carefully advanced until the aortic annulus, positioned high (Figure 3b), and deployed under rapid ventricular pacing (160 beats/min). The deployment was carried out in an extremely slow and careful way, without recapture, in a single attempt (Figure 3c,d). After final deployment, rapid ventricular pacing stimulation was kept (140 beats/min) until delivery system removal, to assure a stable final position (Figure 3e). The final contrast injection showed proper prosthesis expansion, 3–5 mm implantation depth, no central or paravalvular leak, and coronary arteries with adequate flow (Figure 3f). There were no rhythm disturbances, and post-dilation was not required. 

The postoperative course was uneventful, and the patient was discharged home asymptomatic (NYHA functional class I). The control echocardiogram demonstrated a well-functioning bioprosthesis (mean aortic valve gradient 9 mmHg) and no residual AR (Figure 4a,b).

Control CT and transthoracic echocardiogram performed 6 months after the procedure showed no residual AR, and a normo-functioning prosthesis with a mean aortic valve gradient of 10 mmHg (Figure 5a,b). 

## 3. Discussion

Although TAVI was originally designed to address tricuspid aortic valves with severe aortic stenosis, its horizon of use has been progressively expanded to other patients’ populations, such as bicuspid aortic valves, Valve-in-Valve, TAVI-in-TAVI and even AR in high-risk patients. 

### 3.1. Aortic Regurgitation Relevance

Aortic valve regurgitation affects about 13% of patients suffering from isolated native left-sided valvular heart disease. In this population, related symptoms tend to appear later in the disease course, when LV dilatation and systolic dysfunction are already settled, both being risk factors for procedural mortality. In patients with severe AR and LV dysfunction (LVEF < 30%), the annual mortality risk is around 20%, and only 5% of them undergo a surgical intervention [14,15,16].

### 3.2. Intervention in Aortic Regurgitation 

According to the current European and American Guidelines, intervention is indicated in the case of significant AR accompanied by symptoms, decreased left ventricular systolic function, or severe LV dilatation. It is also stated that, for isolated chronic AR, TAVI is challenging because of aortic annulus and aortic root dilation, in addition to lack of sufficient leaflet calcification (Figure 6). Therefore, TAVI is rarely feasible and indicated only for carefully selected patients with a prohibitive surgical risk and in whom valvular calcification and annular size are appropriate for a transcatheter approach [17,18].

### 3.3. Aortic Regurgitation Particularities 

Compared to aortic stenosis, TAVI in PNAR carries many particularities related to anatomical and pathophysiologic factors. While degenerative aortic stenosis is a result of progressive aortic valve leaflets and annulus calcification, PNAR is usually the result of leaflet degeneration or incompetence, aortic root dilatation with aortic annulus enlargement, or both. Besides, severe PNAR is characterized by volume overload and eccentric hypertrophy (instead of pressure overload and concentric hypertrophy, typical of aortic stenosis) associated with LV cavity structural modifications and progressive LV dysfunction [19]. Furthermore, despite being younger, patients with AR are usually referred to treatment in an advanced disease stage, with irreversible LV dysfunction and severe pulmonary hypertension [19,20,21]. 

The absence of annular and leaflet calcification, which is necessary for device anchoring and stabilization, along with an increased stroke volume and aortic root dilatation, makes device positioning and deployment very difficult and predisposes to device embolization or malposition and significant residual AR [22].

Large annuli associated with PNAR is another relevant factor. In a previous study by Yousef A et al., the mean aortic annulus was 24.6 cm, and almost half of the patients required the largest available THV sizes (29 mm and 31 mm). Hence, appropriate sizing can be limited by the availability of large enough prosthetic valves, potentially contributing to a high paravalvular leak (PVL) incidence. Finally, a significant number of patients with PNAR have an elliptical aortic annulus with a significant eccentricity index. Such asymmetric annuli can similarly result in postoperative PVL despite the correct prosthesis positioning [23]. 

### 3.4. TAVI in Aortic Regurgitation

Despite the limitations discussed above, first-in-human reports and cases series have suggested the feasibility and safety of TAVI in AR. These early cases involved the compassioned use of the CoreValve or SAPIEN systems [8,9,10]. In 2012, D’Ancona G et al. reported the use of an oversized 29-mm SAPIEN valve to treat a patient with PNAR who had a long-term left ventricular assist device (LVAD) [8]. After that, Roy et al. published a retrospective analysis of 43 high-risk patients (mean age 75.3 ± 8.8 years, mean logistic EuroSCORE 26.9 ± 17.9%, mean STS score 10.2 ± 5.3%) who had received a CoreValve prosthesis to treat PNAR. As part of the procedure protocol, two pigtail catheters were placed in different Sinuses of Valsalva in order to guide the THV delivery, which was done under rapid ventricular pacing. A 97.7% success rate (according to protocol) was reported and in 18.6% of the cases, a second valve was required during the index procedure due to residual AR. Post-procedure AR grade ≤ I was reported in 79.1%. At 30 days, the major stroke incidence was 4.7%, and the all-cause mortality rate was 9.3%. The 1-year all-cause mortality rate was 21.4% [9].

Following these first cases, several studies have demonstrated improved outcomes, driven mainly by less valve malposition, lower rates of post-procedural AR, and lower cardiovascular mortality, with device success rates ranging from 74.3% to 85% according to Valve Academic Research Consortium (VARC-2) criteria [24,25,26,27]. This is especially true when third-generation devices were chosen compared to older generations.

### 3.5. Results of TAVI for PNAR Comparing First- Versus Newer-Generation Devices

Yousef A et al. conducted a systematic review evaluating 175 patients with PNAR (median age 73.8 years, mean STS score 9.5 ± 3.4%, mean Logistic EuroScore 23.8 ± 4.8%) from 31 studies who had undergone TAVI using a variety of THV (CoreValve, JenaValve, Direct Flow, Acurate TA, J-Valve, SAPIEN, Lotus). Device success, according to the VARC-2 definition, was achieved in 86.3% of patients (8 cases had ≥ moderate PVL, 6 had valve malposition, 3 needed immediate surgical aortic valve replacement, and 1 had late valve embolization). There were no procedural deaths, myocardial infarctions, or annular ruptures. The 30-day outcomes showed a mortality rate of 9.6%, with 11.3% of the cases requiring a second THV, 10.7% needing permanent pacemaker, and 17.7% presenting ≥ moderate PVL. Outcomes with second-generation valves were significantly better compared with first-generation (device success: 78.4% vs. 96.2%, significant residual AR: 8.3% vs. 0.0%, need of a second valve implant: 23.4% vs. 1.7%) [23].

In a larger analysis, Yoon SH et al. reported the results of 331 patients (mean age 74.4 years ± 12.2, mean STS score 6.7 ± 6.7%) with severe AR treated with early- (36%) or newer-generation (64%) devices at 40 centers. Overall device success was 74.3% (61.3% early- vs. 81.1% newer-generation; *p* < 0.001). Second valve implantation (12.7% vs. 24.4%; *p* = 0.007), and moderate to severe post-procedural AR (4.2% vs. 18.8%; *p* < 0.001) were significantly lower with newer-generation devices. There was no significant difference in 1-year all-cause mortality (28.8% vs. 20.6%; *p* = 0.13), however new-generation devices were associated with lower 1-year cardiovascular mortality (9.6% vs. 23.6%; *p* = 0.008). The absence of calcium or the presence of mild calcification was associated with lower device success rate for first-generation devices, but not for second-generation, while a larger annulus (>25.2 mm) was associated with less frequent device success for both first- and second-generation devices. On a multivariable analysis, ≥ moderate post-procedural AR was independently associated with 1-year all-cause mortality [hazard ratio (HR) 2.85; 95% confidence interval (CI) 1.52–5.35; *p* = 0.001). A valve oversizing of at least 15% was associated with less frequent significant residual AR [26].

Similarly, De Backer O et al. reported the early safety and clinical efficacy of TAVI for PNAR in a multicenter (46 centers) study with 254 high-surgical risk patients (mean age 74 ± 12 years, mean STS score 6.6 ± 6.2%) who underwent a TAVI procedure with early- (43%) or newer-generation (57%) devices. The overall device success according to VARC-2 criteria was 67% (47% vs. 82%; *p* < 0.001). Second-generation devices were associated with less valve malpositioning (9% vs. 33%), significant post-procedural AR (4% vs. 31%), and cardiovascular mortality. Interestingly, as part of the study, the authors focused on THV CT-scan sizing and found a significant increase in the incidence of device embolization with relative THV under- or oversizing when compared with neutral sizing [25].

Evaluating the use of self-expanding THV systems (CoreValve = 81; Evolut R = 149) from the STS/ACC TVT Registry, Anwaruddin S et al. described the outcomes of 230 patients (mean age 68.7 ± 15.1 years, mean STS score 8.6 ± 9.1%) with native AR. Device success was 81.7% (CoreValve 72.2% vs. Evolut R 86.9%; *p* = 0.001), with a 30-day all-cause mortality of 13.3%. At 30 days, moderate and severe AR were observed in 9.1% and 1.4% of patients, respectively (CoreValve 19.1% vs. Evolut R 6.3%; *p* = 0.02). On a multivariable analysis, the number of valves used (hazard ratio [HR] 2.361, 95% CI 1.643 −3.391; *p* < 0.001), albumin < 3.3 mg/dL (HR 3.358, 95% CI 1.551–7.273; *p* = 0.002), and LVEF (HR 0.978, 95% CI 0.957–1.000; *p* = 0.047) were associated with 30-day all-cause mortality [28].

In 2020, Takagi H et al. published the results of a systematic review of 11 studies including 911 patients who underwent TAVI for PNAR. Device success (overall: 80.4%; 67.2% vs. 90.2%; *p* < 0.001), moderate or higher PVL (overall: 7.4%; 17.3% vs. 3.4%; *p* < 0.001), 30-day all-cause mortality (overall: 9.5%; 14.7% vs. 6.1%; *p* < 0.001), mid-term (4 months—1 year) all-cause mortality (overall: 18.8%; 32.2% vs. 11.8%; *p* < 0.001), life-threatening/major bleeding complications (overall: 5.7%; 12.4% vs. 3.5%; *p* = 0.015), and major vascular complications (overall: 3.9%; 6.2% vs. 3.0%; *p* = 0.041) were significant better with newer-generation devices. The multivariable analysis showed that > 8% STS, major vascular complications, and ≥ moderate post-procedural AR were independently associated with increased 30-day all-cause mortality, while ≥ moderate baseline mitral regurgitation, ≤ 45% LVEF, >8% STS, ≥ stage 2 acute kidney injury, and ≥ moderate post-procedural AR were independent predictors of 1-year mortality [29].

Most recently, Yin WH et al. also showed a higher success rate (100% vs. 33%; *p* < 0.01), a less frequent need for a second THV implanting (0% vs. 53%; *p* < 0.01), and a better event-free survival for newer- compared with early-generation devices [30]. 

Lastly, Schneeberger Y et al. reported a series of 9 patients with PNAR (mean age 74.4 ± 7.1 years, mean logEuroSCORE II 5.5 ± 3.6%, mean STS 6.2 ± 3.0%) treated with self-expandable Acurate Neo and Neo2 prostheses (Boston Scientific Co., Marlborough, MS, USA). Device success was 100%, early safety 77.7% (7/10), and 30-day mortality 0%. PVL was traced in eight patients (77.7%) and mild in two (22.2%). The majority of the patients were treated under local anesthesia and/or analgosedation (88.8%) [20].

### 3.6. Device selection and Procedural Considerations

Among non-dedicated devices used for PNAR, the self-expanding CoreValve/Evolut has been preferred over balloon-expandable valves due to the possibility of oversizing the prosthesis while preserving a low risk of annular rupture by relying on its radial force at the annular and the ascending aorta level. It can also be recaptured and repositioned, which theoretically makes the prosthesis behave more predictably. Therefore, in patients with a high comorbidity burden who are not eligible for open surgery and coming from centers where dedicated devices are not available (such as our Brazilian center), self-expanding devices have been used as an off-label alternative.

Optimal device selection and valve sizing should be based on careful aortic valve and annulus imaging analysis. The area and the perimeter measurements derive from the diameters obtained by CT or TEE evaluation using the largest annular diameter observed in systole. Published data recommend a 10–20% oversize when selecting the THV size, with the caution to not oversize beyond 20% due to the risk of annular rupture and conduction system abnormalities [31,32,33]. Based on the manufacturer’s recommendations, the percentage diameter oversizing for the CoreValve ranges from 12–30% for the 23-, 26-, and 29-mm sizes, and from 7–19% for the 31-mm device THV [34]. Barbanti et al. demonstrated that >20% annular area oversizing predicted an increased risk of aortic annular injury when earlier-generation balloon-expandable valves were used [35]. In this same line, Maeno Y et al. identified a 7.2% annular area oversizing as the optimal threshold to reduce ≥ mild PVL when balloon-expandable valves were implanted, with aortic annular injuries mostly occurring with >14% oversizing (average of 15.5%). When the SAPIEN 3 is used, since this valve has a covered skirt designed to reduce PVL, the authors suggested that an annular area oversizing between 7% and 15% may be optimal in patients with LV outflow tract calcification [36]. For the Acurate Neo/Neo2, a modified sizing algorithm with an oversizing ratio >10% was proposed by Schneeberger et al. for the AR indication [20]. 

It is important to consider that device dimension is not the only factor implied in the THV embolization risk. The increased stroke volume caused by AR, the low implantation height favored by the absence of fluoroscopic calcific landmarks, and pacing failure are also well-known risk factors. Identifying these risk factors is particularly useful to adopt some precautions after valve implantation, such as a prolonged observation time without removing the wire to avoid valve inversion in case of embolization and to allow subsequent balloon recapture maneuvers [37].

Regarding the implantation technique, it changed substantially since the availability of a retrievable system (CoreValve/Evolut) that allows higher implantations without the fear of valve prolapse in the aorta. Higher implantation allows a more favorable valve oversizing, reducing PVL risk, and minimizing the risk of damaging the atrioventricular conduction system. Furthermore, high valve implantation (not lower than 4–5 mm in the LV) assures a more stable position.

Given the absence of calcification and fluoroscopic landmarks, many operators use two pigtail catheters in different Sinuses of Valsalva, or CT fusion-guided imaging to guide the valve implantation. Balloon predilatation should not be performed unless it is used to measure the annulus. Rapid ventricular pacing is mandatory for balloon-expandable valves, and can also be used with self-expanding valves to reduce stroke volume, stabilize the annulus, and limit THV motion [19,33].

Lastly, it is important to comment that a new balloon-expandable platform has recently become commercially available, the Myval system (Meril Life Sciences Pvt. Ltd., Gujarat, India), which covers larger annulus dimensions (Myval 30.5 mm is indicated to a perimeter up to 95.82 mm, Myvalv 32 mm is indicated to a perimeter up to 100.53 mm). The use of a 32 Myval system to treat a patient with AR and LVAD was newly reported [38].

### 3.7. Dedicated Devices

While non-dedicated devices were adapted from aortic stenosis use, dedicated devices have been developed to be implanted in non-calcified valves anchoring in the aortic annulus and clipping the native valve leaflets. 

The JenaValve was the first dedicated device to get the CE-mark approval for the treatment of AR based on its anatomically correct positioning in the native cusps and clipping of the THV onto the native leaflets. It is a porcine root self-expanding valve on a nitinol frame with three integrated locators and a sealing ring of 24 diamond-shaped cells. The locators align the device with the native leaflet anatomy and act as a strut onto which the nitinol frame is expanded, essentially clipping the device to the native leaflets. This engagement mechanism allows valve anchoring independent of cusp calcification, making it an ideal design for PNAR treatment [12,39]. At the beginning of use, the device was available in 4 sizes (19, 23, 25, 27 mm) with a 21 Fr delivery catheter being used for transapical approach. The transapical implantation of the JenaValve in the earliest 31 patients (age 73.8 ± 9.1 years; mean logistic EuroSCORE 23.6 ± 14.5%**)** with AR was successful in 30 of them (97%), with a 30-day all-cause mortality rate of 12.9% and a 6-months mortality rate of 19.3% [12]. 

The 1-year outcomes of TAVI for PNAR using the JenaValve system in the JUPITER registry were published in 2017. In this study, 30 patients (mean age 74.4 ± 9.3 years, mean STS 4.9 ± 3.5%) were enrolled. Procedural success was 96.7%, with one patient requiring open surgery. No annular rupture or coronary ostia obstruction was reported. The 30-day mortality was 10.0% and the combined safety endpoint was met in 13.3%. PVL was not present or trivial in 84.6% and mild in 15.4%. The rate of permanent pacemaker implantation was 3.8%. One-year all-cause mortality was 20.0%, and cardiovascular mortality was 10.0%. This translates into a 1-year Kaplan–Meier survival of 79.9%, with a 1-year combined efficacy of 73.1%. At 2 years, the mean transvalvular gradient was 9.7 ± 2.9 mmHg [40]. 

Another dedicated second-generation device is the J-Valve, which has a porcine aortic valve within a self-expanding nitinol stent and a unique system composed of three U-shape graspers that facilitate intuitive self-positioning implantation providing axial and radial fixation by embracing the native leaflets. The valve is delivered by an 18-Fr steerable delivery system. The valve cannot be recaptured and is currently available in 5 sizes. It is chosen according to annular sizing, aiming for a 10% oversizing for the AR indication. Once the J-Valve is brought into the ascending aorta, the anchor rings are opened above the native aortic valve and advanced, permitting anatomical self-alignment in the aortic sinuses and clasping of the native valve leaflets. The valve is then deployed, without the need for pacing, within the anchor rings, capturing the native leaflets, aiming for a final position of 70% aortic and 30% ventricular. Successful first-in-human implantation was reported in 2015 [41].

Liu H et al. reported the early results of the transapical use of the J-Valve system in 43 high-risk patients with severe PNAR (mean age 73.9 ± 5.7 years, mean logistic EuroScore 25.5 ± 5.3%). Successful implantation was achieved in 42 patients (97.7%). The 1-year outcomes included all-cause mortality (4.7%), disabling stroke (2.3%), new permanent pacemaker (4.7%), and valve-related reintervention (7.0%). At the 1-year follow-up, post-procedural PVL was none or trace in 30/39 patients and mild in 8/39 patients, and the mean transvalvular gradient was 10.4 ± 4.5 mmHg [42]. In terms of durability, Li F et al. reported the 4-year follow-up of 18 patients (14 with aortic stenosis and 4 with AR) treated with the transapical J-valve system. In patients whose indication was AR, the mean gradient did not increase significantly from discharge to 4-year follow-up, remaining <10 mmHg, with no residual valvular AR or PVL [43].

Since the transfemoral approach has been the preferred route for TAVI implantation, both JenaValve and J-valve were modified and evolved to be deployed through a transfemoral approach [44]. Currently, an early feasibility study of the transfemoral JenaValve is underway in the United States (The ALIGN-AR Trial) [45]. Moreover, the first transfemoral J-valve was successfully implanted in-human in 2019 [46] and patients with AR are being currently enrolled in a compassionate-use study [47].

In the 2022 EuroPCR, Alexander R. Tamm presented data on 45 consecutive high-risk patients (mean age 77 years, mean EuroSCORE II 7.1%) who received a transfemoral JenaValve system at 6 German centers. All patients presented moderate–severe or severe AR, nearly one-third had previously undergone cardiac surgery and more than half (58%) had an LVEF ≤ 50%. Mean procedure time was 77 min. The primary efficacy endpoint, technical success (final mean gradient < 20 mmHg and reduction of more than one AR grade) was achieved in 100% of patients. There was no major or life-threatening bleeding, and the minor bleeding rate was 6.7%. One patient (2.2%) had a major vascular complication. Regarding safety, none of the patients required conversion to open surgery, had a stroke, or died. Nine patients (23%) needed a new pacemaker. At discharge, the mean aortic valve gradient was 4.04 mmHg and the mean aortic valve area was 2.62 cm^2^. Most patients had no or trace PVL (56% and 36%, respectively), while 8.9% had mild [48].

## 4. Conclusions

Since AR prevalence increases with age, a growing number of patients with AR and need for TAVI can be anticipated. Although the surgical approach remains the standard-of-care intervention in patients with severe AR, TAVI has emerged as an option for those at high or inoperable risk.

This case demonstrated the feasibility of TAVI in a very large annulus with PNAR using a non-dedicated self-expanding device. Despite all the particular challenges related to a TAVI in PNAR, some tips and tricks (Figure 7) such as careful preprocedural planning, use of rapid ventricular pacing, placement of two pigtail catheters in different Sinuses of Valsalva, and careful nose cone removal can contribute to procedural success. When available, dedicated devices will contribute to dealing with such challenging anatomies.

## Figures and Tables

**Figure 1 jcm-11-05181-f001:**
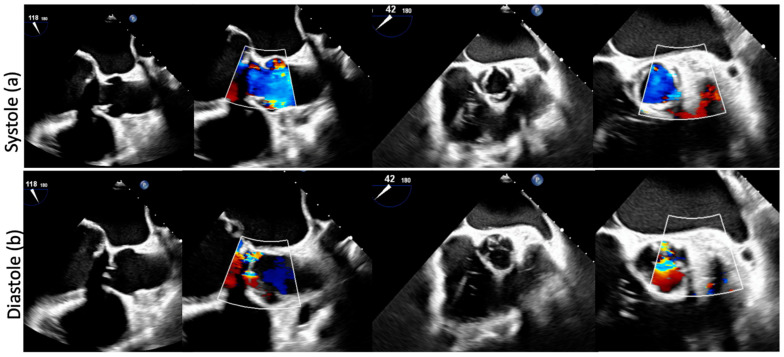
A preoperative transesophageal echocardiogram in systole (**a**) and diastole (**b**) showing annulus dilatation, a large central coaptation defect, severe AR, thin leaflet and absence of annulus and leaflets calcification.

**Figure 2 jcm-11-05181-f002:**
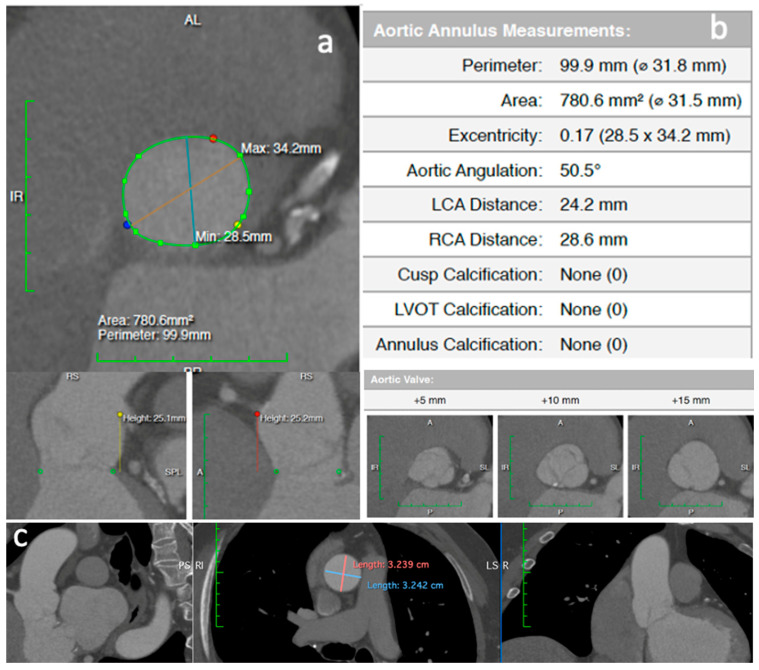
A preoperative computed tomography evaluation (**a**) showing large annulus (area 780 mm^2^, perimeter 99 mm) (**b**) and normal ascending aorta dimensions (**c**).

**Figure 3 jcm-11-05181-f003:**
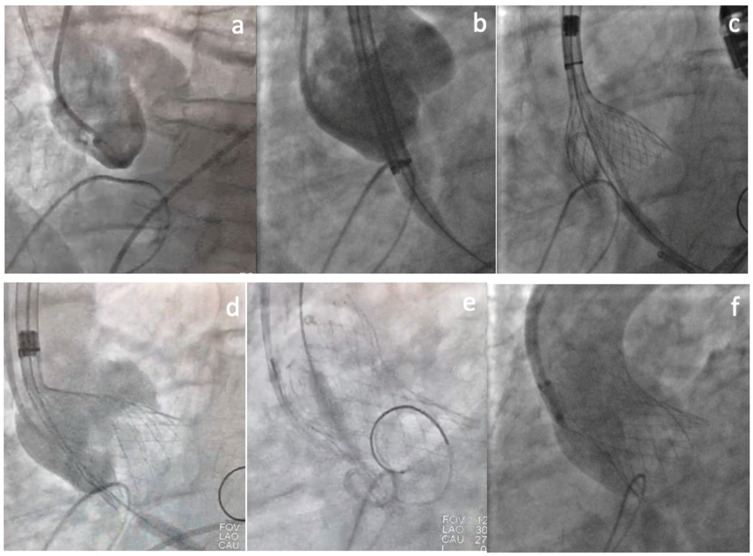
TAVI intraprocedural steps: two pigtail catheters positioned in the aortic Sinuses of Valsalva (**a**); Transcatheter heart valve initial deployment position (**b**); TAVI slow deployment under rapid ventricular pacing (**c**); implant height (3–5 mm depth) (**d**); careful nose cone removal (**e**); TAVI final deployment position (**f**).

**Figure 4 jcm-11-05181-f004:**
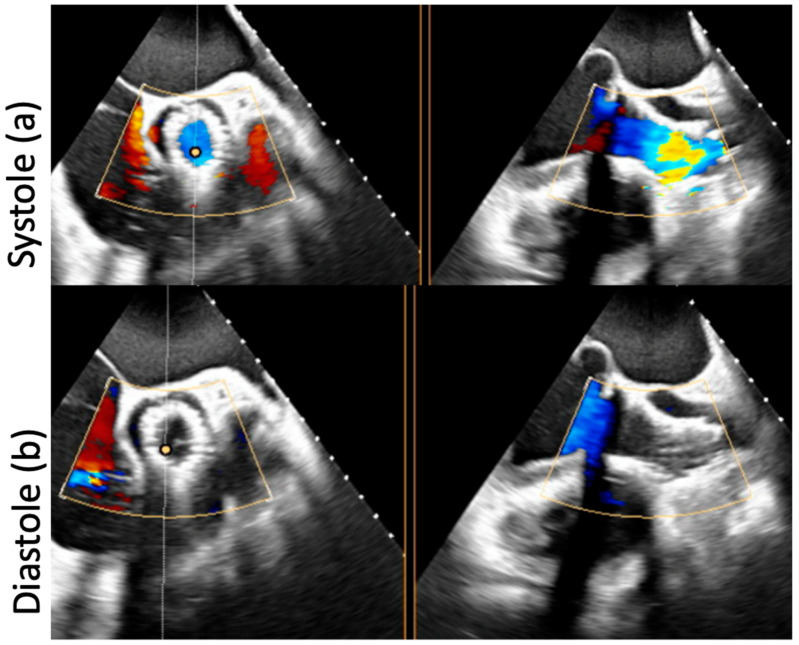
The final control transesophageal echocardiogram in systole (**a**) and diastole (**b**) showing a well-expanded valve with no residual AR.

**Figure 5 jcm-11-05181-f005:**
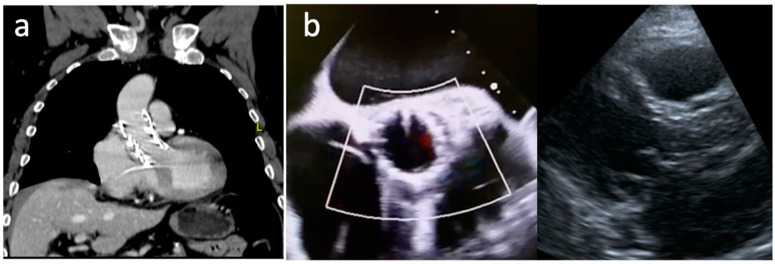
The control computed tomography angiography (**a**) and transthoracic echocardiogram (**b**) showing a well-positioned and normo-functioning THV with no residual AR and a mean gradient of 10 mmHg.

**Figure 6 jcm-11-05181-f006:**
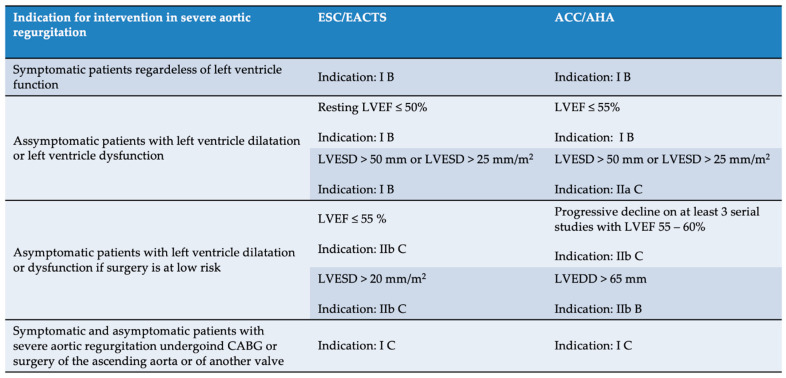
Indications for intervention in significant aortic valve regurgitation according to the European (ESC/EACTS—European Society of Cardiology/European Association for Cardio-Thoracic Surgery) and American Guidelines (ACC/AHA—American College of Cardiology/American Heart Association). LVESD—left ventricle end-systolic diameter; LVEF—left ventricle ejection fraction; LVEDD—left ventricle end-diastolic diameter; CABG—coronary artery bypass graft [17,18].

**Figure 7 jcm-11-05181-f007:**
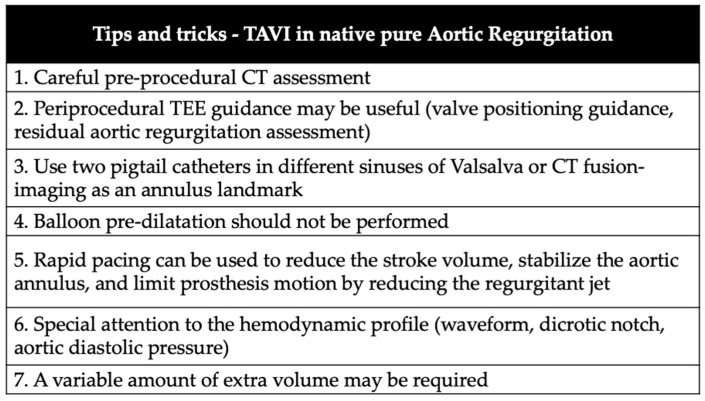
Tips and tricks to perform a TAVI in PNAR.

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
