# Peer review of "Transcatheter Aortic Valve Implantation for Pure Native Aortic Regurgitation: The Last Frontier"

_jcm, 2022, doi:10.3390/jcm11175181_

Round 1
Reviewer 1 Report
The authors present a successful case of Evolut 34mm in PNAR, along with a review of the literature.
The focus should be on technical aspects of TAVI in PNAR including device selection and size, as such oversizing is different among the various devices (balloon expandable should be oversized to approximately 10%, while self expanding valves can be oversized over 25 and even 30%), moreover calcification is key for device selection as lack of it prohibits the use of BE THV. These points are shortly mentioned in the final part of the discussion, but I suggest to shorten the review and summarize its main points while elaborating on the technical aspects - sizing, pacing during deployment, pacing after deployment (more controversial) etc.
Author Response
Dear Editors and Reviewers,
We appreciate your comments and questions, which help to improve the quality of our articles.
We have added a new section discussing all aspects of prosthesis choice and the procedural steps.

Reviewer 2 Report
The manuscript by Tagliari and colleagues presents a case of TAVI in aortic regurgitation without significant sclerotic valve degeneration as basis for a review article summarizing current knowledge in the field. In general, the article is well structured, easy to follow and excellently written. Since it addresses a remaining challenge in the treatment of elderly high-risk patients with PNAR, the educational value is high. Nevertheless, there are some minor points, which should be addressed by the authors before publication:
1. Case presentation:
- Please state on the maximum diameter of the ascending aorta measured by CT.
- What is the suspected origin of the central coaptation defect resulting in severe aortic regurgitation is there a status post aortic valve manipulation?
- Please give also the STS-Score in addition to Euro Score II.
- What was the rationale to use a self-expanding prosthesis instead of a balloon expandable valve? Since for both, the Evolut R 34mm as well as the Edwards Sapien 3 29mm, the annulus diameters are out of range, one could also have discussed to use an over-expanded ES3 device, which might gain more stability in a non-calcified annulus. Please comment.
- It is not clear to the reviewer why you performed rapid stimulation for 10 minutes after deployment. The explanation given in the text is not convincing. Please revise.
2. Review article
- Page 8 line 265-278: Please add a short statement of the availability of the JenaValve in your country because, without doubt, this would have been the device of choice in PNAR.
- Figure 7: Please rearrange the layout, which is not very attractive in its current form.
- Figure 8, point 2: TEE might not be mandatory in PNAR-TAVI procedures, especially against the background of deep sedation and longer procedural times. Positioning can be probably done by angiography and residual AR / PVL can be also seen by trans-thoracic echocardiography. Please attenuate this recommendation a little bit.
Author Response
Dear Editors and Reviewers,
We appreciate your comments and questions, which help to improve the quality of our articles.
Below we address each one of the points raised by the reviewers.
- Please state on the maximum diameter of the ascending aorta measured by CT.
The maximum ascending aorta diameter was 32 mm (A figure showing it was added).
- What is the suspected origin of the central coaptation defect resulting in severe aortic regurgitation is there a status post aortic valve manipulation?
The most probable origin was annulus dilatation. We do not have an early post-mitral valve replacement echo to evaluate what were the annulus dimensions immediately after this procedure.
- Please give also the STS-Score in addition to Euro Score II.
The STS-Score was added (4%, STS does not contemplate hostile chest).
- What was the rationale to use a self-expanding prosthesis instead of a balloon-expandable valve? Since for both, the Evolut R 34mm as well as the Edwards Sapien 3 29mm, the annulus diameters are out of range, one could also have discussed to use an over-expanded ES3 device, which might gain more stability in a non-calcified annulus. Please comment.
Since we did not have a large experience with TAVI in pure native aortic regurgitation before this case, we decided to use the prosthesis with the larger clinical experience according to the current literature, which is the CoreValve/Evolut system. Besides, the idea of using a repositionable and retrievable THV also influenced our choice. We added a paragraph discussing the prosthesis choice.
"Among non-dedicated devices used for PNAR, the self-expanding CoreValve/Evolut has been preferred over the balloon-expandable valves due to the possibility of oversize the prosthesis while preserving a low risk of annular rupture through relying on its radial force at both the annular level and ascending aorta. It can also be recaptured and repositioned, which theoretically makes the prosthesis behave more predictably. Therefore, in patients with a high comorbidity burden who are not eligible for open surgery and coming from centers where dedicated devices are not available (such as our Brazilian center), self-expanding devices have been used as an off-label alternative".
- It is not clear to the reviewer why you performed rapid stimulation for 10 minutes after deployment. The explanation given in the text is not convincing. Please revise.
We decided to keep rapid pacing stimulation to decrease stroke volume during the immediate period post valve deployment while we checked the valve position and removed the guide wire and nose cone. There is no formal recommendation to do this, but I believed (empirically) that this stroke volume reduction could contribute to prosthesis stabilization.
- Review article
- Page 8 line 265-278: Please add a short statement of the availability of the JenaValve in your country because, without doubt, this would have been the device of choice in PNAR.
Unfortunately, JenaValve is not available in Brazil. If it had been available, it would have been the choice for sure.
- Figure 7: Please rearrange the layout, which is not very attractive in its current form.
This figure was excluded.
- Figure 8, point 2: TEE might not be mandatory in PNAR-TAVI procedures, especially against the background of deep sedation and longer procedural times. Positioning can be probably done by angiography and residual AR / PVL can be also seen by trans-thoracic echocardiography. Please attenuate this recommendation a little bit.
This figure was modified.
